# High Prevalence of Recombinant Porcine Endogenous Retroviruses (PERV-A/Cs) in Minipigs: A Review on Origin and Presence

**DOI:** 10.3390/v13091869

**Published:** 2021-09-18

**Authors:** Joachim Denner, Hendrik Jan Schuurman

**Affiliations:** 1Institut für Virologie, Freie Universität Berlin, 14163 Berlin, Germany; 2Schubiomed Consultancy, 3583 VH Utrecht, The Netherlands; schubiomed@gmail.com

**Keywords:** porcine endogenous retroviruses, recombination, xenotransplantation, PERV-A/C recombination, minipigs, miniature swine, recombination in vitro, recombination during life

## Abstract

Minipigs play an important role in biomedical research and they have also been used as donor animals for preclinical xenotransplantations. Since zoonotic microorganisms including viruses can be transmitted when pig cells, tissues or organs are transplanted, virus safety is an important feature in xenotransplantation. Whereas most porcine viruses can be eliminated from pig herds by different strategies, this is not possible for porcine endogenous retroviruses (PERVs). PERVs are integrated in the genome of pigs and some of them release infectious particles able to infect human cells. Whereas PERV-A and PERV-B are present in all pigs and can infect cells from humans and other species, PERV-C is present in most, but not all pigs and infects only pig cells. Recombinant viruses between PERV-A and PERV-C have been found in some pigs; these recombinants infect human cells and are characterized by high replication rates. PERV-A/C recombinants have been found mainly in minipigs of different origin. The possible reasons of this high prevalence of PERV-A/C in minipigs, including inbreeding and higher numbers and expression of replication-competent PERV-C in these animals, are discussed in this review. Based on these data, it is highly recommended to use only pig donors in clinical xenotransplantation that are negative for PERV-C.

## 1. Introduction

Worldwide several breeds of minipigs (also called miniature swine or micropigs) have been established (Table 1) and most of these minipigs are used in biomedical research and development, only some of them are favored as pets. The main fields in biomedicine using minipigs are basic research [1], experimental surgery [2], pulmonology [3], cardiology [4], transplantation [5], toxicology [6], pharmacology and safety assessment of new drugs [7], orthopedic procedures [8], aging studies [9], medical devices [10], food additives [11] and agrochemicals [12]. The skin, gastrointestinal tract, cardiovascular system, kidney and bladder are very similar to humans in terms of anatomy and function, e.g., their liver metabolism is similar to humans for most cytochrome P450 (CYP) isoenzymes [13]. This allows performing pharmacological and toxicological studies that are acceptable in regulatory filings [14]. Now more than a decade ago, the EU-sponsored RETHINK project has highlighted this use of minipigs in biomedicine, including toxicity testing, which as a consequence adds to the 3Rs (replacement, refinement and reduction) of animal testing [15].

The most known minipigs are the Minnesota minipigs from the Hormel Institute in the United States [35], the Massachusetts General Hospital swine leukocyte antigen (SLA)-defined miniature pigs [36], the Göttingen minipigs [13], the Aachen minipigs [31,37], the Munich miniature swine [33,38], the Yucatanmicro pigs [39,40,41], the Chinese bama minipigs [42], the Brazilian minipig [1] and the Mini-LEWE [43]. The Minnesota miniature swine were the first minipigs developed in the USA in 1940s [44] and they are the basis of most if not all other minipig breeds, including the Göttingen minipigs, which were the first miniature pig breed developed in Europe.

Most of the minipigs used in biomedical research are bred under clean conditions, for example, at Ellegaard Göttingen Minipigs A/S, Denmark, the Ellegaard-Göttingen minipigs are bred in biosafety barriers under specified pathogen-free (SPF) conditions: twice a year there is monitoring for numerous microorganisms including 27 bacteria, 16 viruses, 3 fungi and 4 parasites [13]. In additional studies, these animals have been tested for altogether 89 bacteria, viruses, fungi and other microorganisms [25,26]. Recently, these minipigs have been used as donors of islet cells in a preclinical study in non-human primates, in which transspecies transmission of porcine viruses could not be documented [27]. Amongst others this study is the basis for a clinical trial with macroencapsulated islet cells from Ellegaard-Göttingen minipigs planned in Germany in the near future.

In this review, we will concentrate on the characterization of porcine endogenous retroviruses (PERVs) in different minipig breeds and on the observation that the recombinant PERV-A/C type is very common in such breeds.

## 2. Data on PERV-A/C Recombinants in Various Minipig Strains

PERVs are integrated in the genome of pigs. PERV-A and PERV-B are in the genome of all pigs and they are polytropic viruses, e.g., they infect cells from different species including humans [45]. Two receptors for PERV-A on human cells are known: the human porcine endogenous retrovirus-A receptor 1 (huPAR1) and the human porcine endogenous retrovirus-A receptor 2 (huPAR2) [46]. Receptors for PERV-B and PERV-C have not yet been identified. PERV-C is present in most but not all pigs and is an ecotropic virus infecting only pig cells [47]. Pig cell lines such as the pig kidney cell line PK15 [48,49,50,51], permanent cell lines from leukemic pigs [51,52,53,54,55] and also primary pig endothelial cells [56] release virus particles. PERVs are also released spontaneously from cultured porcine bone marrow cells in culture; remarkably, combinations of porcine hematopoietic cytokines such as interleukin-3, granulocyte-macrophage colony-stimulating factor and stem cell factor had no additional effect on this release [57]. Virus particles released from PK15 cells are able to infect human cells in vitro [51] and all clones of replication-competent PERV-A and PERV-B have been isolated from PK15 cells or cell lines infected by viruses released from PK15 cells [58,59]. In contrast, particles released from the bone marrow infected only pig cells in vitro [57].

### 2.1. PERV-A/C Recombinants in NIH and Yucatan Miniature Pigs

Studies on National Institute of Health (NIH) miniature pigs or Yucatan pigs maintained by the NIH animal facility at Poolesville, MD showed that their peripheral blood mononuclear cells (PBMCs) upon stimulation with the T-lymphocyte mitogen phytohemagglutinin (PHA) manifested a peak of PERV release at day 5 of cultivation [16]. When the supernatants of the stimulated pig PBMCs were co-cultivated with cells of the human embryonic kidney 293 cell line, transmission of PERV was observed [16]. Cells of the human 293 cell line have been shown to be highly susceptible for PERV infections due to the absence of restriction factors [60]: these cells are, therefore, used because of their propensity for transmission, e.g., in biotechnology industry to produce therapeutic proteins and viruses for gene therapy.

The origins of the NIH miniature pig and the Yucatan pig are not very well understood. The Yucatan minipig is derived from a herd living in the wild in the Yucatan peninsula of Mexico [39,40,41]. The NIH miniature pig is derived from crosses, which included both feral and domesticated breeds found within the United States and elsewhere [35,36,61].

Sequencing the virus isolated from the mitogen-stimulated PBMCs of the NIH minipig revealed the presence of a recombinant between PERV-A and PERV-C, one breakpoint being in the transmembrane envelope protein, the other upstream to the env [17]. Due to this recombination, the backbone PERV-C acquired the receptor binding site of the human-tropic PERV-A (Figure 1), which allowed the virus to infect human cells using receptors mentioned above. At that time, it was not clear whether the recombination happened in the pig itself or during the in vitro infection experiment. Noteworthy, the recombinant was not present in the germ line (see below).

Upon the quick passage of the PERV-NIH, i.e., the PERV-A/C isolated from the NIH miniature minipigs, on human 293 cells, the quick speed causing selection pressure, higher titers of the virus as well as longer long terminal repeat (LTR) sequences were observed [17,62]. The longer LTRs were due to a multimerization of repeats within the LTR, which functions as binding sites of transcription factors. A similar multimerization of repeats was observed when a cell-free PERV-A from PK15 cells was passaged on human 293 cells [62]. However, when the passaging of the PERV-A/C was stopped, i.e., there was no more a selection pressure, the virus lost some of the repeats and acquired mutations in the transcription factor binding domains reversing their function [63]. This in vitro evolution experiment simulated the situation that might have happened in the natural evolution of PERV [64]. In a parallel experiment, passaging molecular clones of PERV-B in 293 cells resulted in the loss and gain of 39-base pair repeats [65].

It is interesting that a PERV-A/C recombinant was released from mitogen-stimulated PBMCs of NIH miniature swine [16], but not from their bone marrow cells stimulated with porcine hematopoietic cytokines [57]. Either the protected environment of the bone marrow, the promiscuity of the PBMCs or the type of stimulation may be the reason of this difference.

PERV-A/C was also found in a commercially available Yucatan micropig herd [32]. Recombinant PERV-A/C was detected integrated in the cellular DNA in some organs like kidney, liver, brain and spleen, with the highest copy number in the kidney. There were no detectable PERV-A/C recombinants in other tissues, indicating that PERV-A/C recombinants were not present in the germ line. PERV protein expression was observed in several organs of the animal using immunohistochemistry and specific antibodies against PERV proteins. Release of reverse transcriptase (RT) activity into the supernatant of mitogen-stimulated PBMCs has been documented for some Yucatan micropigs, but not for others [66]. High RT activity in the supernatant indicates release of virus particles. This method as well as the described immunohistochemistry cannot discriminate between PERV-A/C and other PERVs; however, in the pelleted virus particles from the supernatant of stimulated PBMCs, RNA from PERV-A, PERV-B and PERV-C were found [66].

### 2.2. PERV-A/C Recombinants in MGH Miniature Swines

For the well-known inbred herd of miniature swine at the Massachusetts General Hospital (MGH) of Harvard Medical School in Boston, MA, infection of human 293 cells was observed after stimulation of PBMCs with PHA and phorbol 12-myristate-13-acetate [18]. The MGH miniature swine and the NIH miniature pig have a common origin [36]. PBMCs from pigs with different SLA haplotypes showed differences in the release of human-tropic and pig-tropic PERVs (target cells human 293 cells and porcine ST-IOWA cells, respectively: the ST-IOWA cell line is used as target because it is PERV-C-negative). The frequency of release and transmission to human cells varied markedly, for example the SLA^d/d^ and SLA^g/g^ lines showed low transmission and at least one animal with other SLA haplotypes (a/a, c/c, h/h and k/k) also transmitted PERV. Irrespective of SLA haplotype, animals contained PERV-C in their genome and released particles able to infect pig ST-IOWA cells. The viruses infecting 293 cells were PERV-A/C recombinants including PERV-A 14/220, which was obtained from an animal of SLA^c/c^ haplotype [18]. The PERV-A pol-env sequence in the genome of PERV-A 14/220 has a length of 850 bp [19]. The recombinant PERV-A 14/220 infected several cell lines from different species at higher titers than the paternal PERV-A derived from PK15 cells [18]. More importantly, such recombinant PERV-A/Cs have not been detected in the germ line of the respective miniature swines [19] and they are not transmitted to the progeny. The recombinant PERV-A/Cs are the product of expression (possibly also replication) and recombination of PERV-A and PERV-C (Figure 1) and de novo integration of the recombinant virus into the genome of somatic cell in the living animals. This is supported by the presence of PERV-A/C recombinants in some but not all organs of various pigs [20,30,32,67].

Recombinant PERV-A 14/220 was found to be 500-fold more infectious than PERV-A [21]. Two determinants of high titer in this virus were identified in a study using a series of chimeric env genes, one was the isoleucine-to-valine substitution in position 140 and the other was in the proline-rich region of the env gene [21]. It was suggested that these mutations may enhance the infectivity by stabilization of the envelope glycoprotein or by increased receptor binding [21].

### 2.3. PERV-A/C Recombinants in Göttingen Minipigs

PBMCs from minipigs at Ellegaard Göttingen Minipigs A/S were tested for PERV-A/C and no recombinants were found [28]. However, PBMCs from Göttingen minipigs at the University of Göttingen, being a source of the minipigs at Ellegaard Göttingen Minipigs A/S, manifested PERV-A/C in 3 out of 11 animals: in one case, the release of virus particles infecting human 293 cells was observed (Figure 1) [30]. The presence of PERV-A/C in freshly isolated PBMC clearly demonstrates that the virus was already present in the living animal. However, in both studies PERV-A/C was not integrated in the germ line of these animals [28,30].

### 2.4. PERV-A/C Recombinants in Aachen Minipigs

Aachen minipigs originated from cross-breeding minipigs at different minipig farms located mainly in Eastern Germany, which started in 1973 [31,37]. PERV-C was found in all investigated animals and PERV-A/C was found in the spleen and liver of 3 out of 13 animals. However, PERV-A/C was not found in PBMCs. These data confirm the results in other breeds that PERV-A/C is not present in the germ line [31].

### 2.5. Chinese Bama Minipigs

PERVs were integrated in the genomes of Chinese Bama minipigs at different copy numbers and the copy numbers of the PERV-C subtype were mostly low [68]. In contrast to the Wuzhishan minipigs, which are all PERV-C negative, 15 out of 20 Bama minipigs carried PERV-C [69]. Unfortunately, there are no reports on PERV-A/C recombinant viruses in these pigs.

### 2.6. PERV-A/C in Other Pig Breeds Than Minipigs

An extensive testing for PERV-A/C was not performed in other swine breeds than minipigs. In some cases, such a testing was performed and the results were negative: examples are wild boars [70,71] and Large White animals [71]. Further studies have been conducted in genetically modified animals generated for xenotransplantation purpose: PERV-A/C recombinants were only found in animals with a minipig proportion in their genome, e.g., in Large White/Duroc/Minipigs (15 out of 28 animals found positive) and in Large White/German Landrace/Duroc/Minipigs (three of five animals found positive) [71]. The minipigs used for crossing in were Berliner minipigs/Mini LEWE.

In a related study, a few PERV-C positive animals were documented in a large number of Landrace x Yorkshire crossbred pigs: however, PERV-A/C was not analyzed in this study [72]. In another study, Large White–Yorkshire × Landrace F1 hybrid animals were analyzed and most of the animals were PERV-C positive: but also in this study PERV-A/C was not analyzed [73].

The only exception describing PERV-A/C in other pig breeds than minipigs is the report on diseased animals in US farms [34]. The animals in herds for commercial swine production included varying proportions of Duroc, Landrace, Yorkshire and Large White genetics and manifested various clinical conditions including diarrhea, wasting and respiratory diseases. The study revealed a much higher prevalence of PERV-A/C recombinants than in healthy animals (25% versus 8.3%, respectively). Interestingly, regarding age PERV-A/C was more prevalent in younger pigs (3–9 weeks of age). Unfortunately, it was not studied in which cells PERV-A/C was present: nevertheless, this result is in line with the observations mentioned above in PBMCs that PERV-A/C expression needs stimulation like that by mitogens [16,66,71]: this mitogen stimulation simulates stimulation of cells of the immune systems in diseased animals.

### 2.7. Differences in the Location of the Recombination Sites

The result of an PERV-A/C recombination (Figure 1A) is the acquisition of the receptor binding site of PERV-A into the sequence of PERV-C, this allows PERV-A/C to infect human cells and cells of other species, whereas PERV-C is infecting only pig cells. Important is that the recombination sites are all different (Figure 1B), indicating that recombination can occur in many places in the env gene, no apparent hot spots were found [19]. The recombination site in the env gene is well studied (Figure 1B), the recombination site upstream env is largely unknown. In addition, a higher replication competence was acquired. Obviously, there are multiple determinants for a difference in titers between PERV-A and PERV-A/C [19,21].

## 3. Possible Reasons for the High Prevalence of PERV-A/C in Minipigs and Its Implications

Recombination frequencies depend on the number of integrated proviruses and on the transcription of the genomes in virus-producing cells. In this context it is noteworthy that PERV-A proviruses are present at higher levels than PERV-B in most pig breeds including minipigs [19,74]: this explains that until now no PERV-B/C recombinants have been identified. In addition, PERV-A RNA transcripts appear to be more abundant than those of PERV-C in cultures of minipig PBMCs [18]. Minipigs may be unique as they possess more copies of PERV-C sequences than some if not most other pig breeds, e.g., Large White pigs [18,22,51,74,75]. This obviously is relevant, as it has been suggested that the level of one or more PERV-C loci and their expression drives the production of PERV-A/C recombinants [20]. Genomic analysis of a MGH miniature swine capable to release PERV-A/C and to infect human cells identified a PERV-C provirus in a region with homology to sequences located on chromosome 7. A PCR amplification assay with specific primers revealed that only 2 out of 5 miniature swine that are not able to transmit PERV in vitro to human or pig cells retained this locus: in contrast, this locus was found in all 5 animals able to transmit recombinant PERV-A/C to both human and pig cells and in all 7 out of 7 animals able to transmit non-recombinant PERV to pig cells [22]. Unfortunately, the exact PERV and PERV-C copy numbers have not yet been documented in most minipigs (Table 1).

It is obvious that in all these recombinant PERV-A/Cs described above the recombination site is different (Figure 1B) [17,18,19,30,62]. It is, therefore, most likely that these PERV-A/C recombinants, which are genetically distinct from each other, have arisen in individual pigs during life: this aside, in all cases a tropism for human cells and a higher replication competence was acquired. The finding that PERV-A/C recombinants were found in some but not all organs of living pigs [23,24,30,33] and also that PERV-A/C recombinant env sequences were found in native porcine PBMCs [24,30], suggests that it is unlikely that the recombination between PERV-A and PERV-C happens vitro, i.e., in the co-culture of pig PBMCs with human 293 cells: it is more likely that this happens in the animal during life.

### 3.1. Knowledge from Inbred Laboratory Mouse Strains, Applicable to the Swine Species

Inbreeding of the minipigs may be one reason for increased PERV copy numbers, especially for the increased numbers of the ecotropic PERV-C that represents the most recent stage in PERV evolution [76]. This phenomenon is illustrated in more detail in the mouse species. In numerous inbred laboratory mouse strains inbreeding has shown to increase the copy number of murine endogenous retroviruses [77,78,79]. Interestingly, poly-tropic mouse endogenous retroviruses (P-MLV) can amplify in the presence of infection by ecotropic mouse endogenous retroviruses (E-MLV), because P-MLV genomes are preferentially packaged in E-MLV particles that then use the E-MLV receptor to enter cells [80,81]. De novo insertions of endogenous retroviruses are estimated to cause 10% of spontaneous mutations in mice [82] and polymorphisms in endogenous retroviruses have been associated with changes in gene expression, especially in genes with differential expression across strains [83,84].

### 3.2. Knowledge from the Feline Species, Applicable to the Swine Species

PERV-A/C viruses were easily detected because of their increased virulence. This is a general feature of recombinant retroviruses and was well described for recombinant feline leukemia viruses (FeLV), avian leukosis viruses (ALV) and koala retrovirus (KoRV). In lymphomagenesis by murine and feline retroviruses, the acquisition of endogenous env sequences is often observed [85]. For example, feline leukemia virus A (FeLV-A) is a naturally occurring retrovirus of the cat, being horizontally transmissible from cat-to-cat. Its malignant potential depends on (i) the regulatory sequences in the LTR, (ii) the cell tropism depending on the surface envelope protein, (iii) the activation of oncogenes by insertional mutagenesis [86] and (iv) the virus load defining the severity of virus-induced immunosuppression [87]. For FeLV also repeat sequences inside the LTR have been identified, which may multimerize and therefore enhance the pathogenicity of the virus. Thymic lymphomas induced by FeLV are usually associated with LTRs containing duplicate enhancers with repeat lengths varying from 39 to 77 base pairs [86]. These repeats in LTR represent in both PERV and FeLV binding sites for transcription factors [62,86]. FeLV-B arises de novo within the infected cat by mutation of FeLV in the surface envelope protein or by recombination between FeLV-A and endogenous FeLV-related elements [86,88]. FeLV-A exhibits like PERV-C an ecotropic host range and uses a thiamine transporter, FeTHTR1, as receptor. FeLV-B is polytropic like PERV-A and PERV-B and uses a different receptor, the phosphate transporters FePiT1 and FePiT2 [89].

### 3.3. Knowledge from Avian Species, Applicable to the Swine Species

ALV can induce various tumors and cause serious economic loss in the poultry industry. ALVs isolated from chickens were divided into six subgroups (A-J). New ALV-K viruses were isolated from chicken in China and were identified as recombinant viruses, either as ALV-K and ALV-E recombinants, or ALV-K, ALV-E and ALV-J multiple recombinant strains containing the U3 region of ALV-J. The viremia and viral shedding level were significantly increased and the viruses could invade and injure the brain tissue of chickens [90]. Another example of how ALV continues evolving to obtain new genomic characters to enhance its pathogenicity is a new ALV-J strain [2]. This virus is highly pathogenic, inducing tumors in multiple organs including bone, liver, spleen and kidney. In the viral genome recombination events with endogenous virus ev-1 sequences were identified and major recombination sites of the genome with were located in 5′ UTR-gag and 3′ UTR regions [91].

### 3.4. Knowledge from Studies in Koalas, Applicable to the Swine Species

The situation in koalas is similar to that in mice and cats. Koala retrovirus type A (KoRV-A) is an endogenous virus, which can recombine with endogenous viral sequences: the result is an exogenous KoRV-B and other KoRVs (C to I), which use another receptor and are more pathogenic, i.e., their specifically associates with fatal lymphoma and leukemia which does not occur for KoRV-A [92].

## 4. Conclusions

Recombination between retroviruses may lead to recombinants which use another receptor and have an increased pathogenicity, or, in the case of PERV-A/C, at least a higher replication rate. It is important to note, that studies on all preclinical (i.e., pig-to-nonhuman primate) and clinical pig-to-human xenotransplantation trials using cells or organs from pigs, have till now not been able to demonstrate trans-species PERV transmission to the recipients. This was also true in the animal model, in which islet cells from PERV-C-positive Göttingen minipigs from Ellegaard Göttingen Minipigs A/S were transplanted into healthy, non-diabetic cynomolgus monkeys using a macroencapsulation device [27].

PERV-A/C recombinants are able to infect human cells and are characterized by a high replication capacity. These recombinants are regularly found in minipigs of different origin. Possible reasons of this high prevalence may be inbreeding, an increased number of PERV-C copies in comparison with other pig breeds and highly active PERV-C sequences. Therefore, for clinical xenotransplantation to be safe, e.g., to prevent transmission of (high-titer) PERV-A/C to the recipients, it is highly recommended to use PERV-C-free donor pigs because such pigs are unable to generate such recombinants.

## Figures and Tables

**Figure 1 viruses-13-01869-f001:**
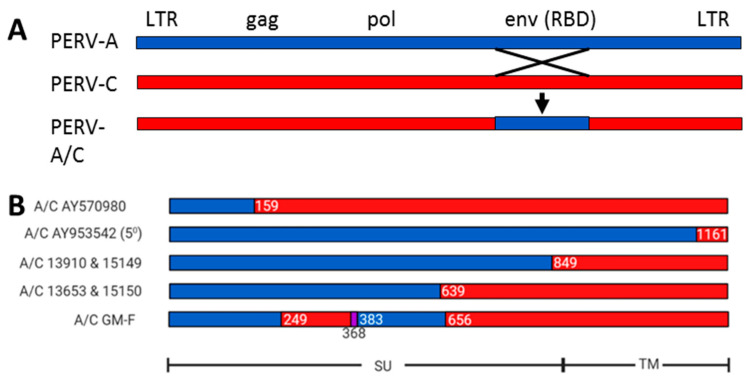
Schematic presentation of the recombination points in PERV-A/C. (**A**), Localization of the retroviral genes gag (group-specific antigen), encoding the core proteins, pol (polymerase), encoding the reverse transcriptase, env (envelope), encoding the envelope proteins with the receptor binding domain (RBD), as well as the long terminal repeats (LTR) of the integrated provirus. The recombination sites in the envelope and upstream to the env are shown. (**B**), Examples of recombination sites in the env gene of several PERV-A/C, AY570980 PERV−A 14/220, Bartosch et al. [19], AY953542 PERV−5^0^, Denner et al., [62], 13910, 13653, 15149, 15159, recombinant PERV-A/C from MGH miniature swines, Martin et al., [23]. A/C GM-F is a recombinant PERV-A/C isolated recently from the Göttingen minipig F [30]. The borders of the surface envelope (SU) and transmembrane envelope (TM) proteins are indicated. Blue indicated PERV−A, red PERV−B, purple is a 15 nucleotide (nt) insert in A/C GM-F, the numbers indicate the last nt of the recombination point.

**Table 1 viruses-13-01869-t001:** Pig breeds with reported PERV-A/C prevalence.

Minipig, Pig Breed	Institution/Company	Characterization	PERV Copy Number	Prevalence of PERV-C	Detection of PERV-A/C	References
National Institutes of Health (NIH) minipigs, Yucatan pigs	NIH animal facility at Poolesville, MD		n.t. ^1^	n.t.	Isolation and characterization of PERV-A/C	[16,17]
Massachusetts General Hospital SLA-defined miniature pigs	Massachusetts General Hospital, Harvard Medical School, Boston, MA	Strong expression of PERV-C correlated with an ability of the PBMC to transmit PERV-A/C recombinants	n.t.	n.t.	Multiple isolation and characterization of PERV-A/C	[18,19,20,21,22,23,24]
Göttingen minipigs	Ellegaard Göttingen Minipigs A/S, Denmark		64 (47–93)	100% (15 animals)	None in PBMCs, organs not tested	[25,26,27,28,29]
Göttingen minipigs	University Göttingen, Göttingen, Germany		77	100% (11 animals)	3/13 in mitogen-stimulated PBMCs, 1 animal released virus	[30]
Aachen minipigs	Aachen Minipig, Heinsberg, Germany		69 (34–97)	100% (13 animals)	3/13 animals in liver, spleen, but not in PBMCs	[29,31]
Yucatan micropig	Charles River, Saint-Aubin-Les-Elbeuf, France		n.t.	1 animal	Proviruses in kidney, liver, brain and spleen	[32]
Munich miniature swine (MMS) of the strain Troll	Institute of Veterinary Pathology, University of Munich	Melanoma	n.t.	100% (5 animals)	2/5 animals in the spleen	[33]
Farm animals, Crossbred pigs (varying proportions of Duroc, Landrace, Yorkshire and Large White genetics)	US swine operations located in Iowa, Kansas, Michigan and North Carolina	Healthy pigs (*n* = 60) and pen-mates with various clinical conditions including diarrhoea, wasting and respiratory disease (*n* = 60).	n.t.	24.5% (89/369 animals)	18.7% (69/369) in serum; 8.3% (5/60) in healthy pigs; 25% (15/60) in clinically affected pigs	[34]

^1^ n.t., not tested.

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
