# Peer review of "High Prevalence of Recombinant Porcine Endogenous Retroviruses (PERV-A/Cs) in Minipigs: A Review on Origin and Presence"

_viruses, 2021, doi:10.3390/v13091869_

Round 1

Reviewer 1 Report

Line 29-32. If you can, please give references for some of these fields or all.

Table 1: the prevalence of Yucatan micropig is 100 % after testing one animal.  Remove the 100 % since the tested animal is only one.

L40 : instead of the “best-known” use “ the most popular or the most known “

L47 49 : For your statement “ The Minnesota miniature swine were the first minipigs developed in the USA in 1940s”  you can give credit or  reference to American Mini Pigs association website: https://americanminipigassociation.com/mini-pig-education/what-is-an-american-mini-pig

L87-91 you said “Cells of the human 293 cell line have been shown to be highly susceptible for PERV infections due to the absence of restriction factors [34]” The referred paper (reference # 34) is not related to PERV and 293. It is related with 293T and HIV-1. Please find the appropriate reference or delete the sentence.

L257-258: you said “ For FeLV also repeat sequences in the LTR have been identified”.  The statement is not clear or does not flow with the preceding one .It is obvious that all retroviruses’ LTRs have repeat sequences.

Author Response

Reviewer 1

Comment 1

Line 29-32. If you can, please give references for some of these fields or all.

Answer 1

References for each of these fields were added.

The main fields in biomedicine using minipigs are basic research [1], experimental surgery [2], pulmonology [3], cardiology [4], transplantation [5], toxicology [6], pharmacology and safety assessment of new drugs [7], orthopedic procedures [8], aging studies [9], medical devices [10], food additives [11] and agrochemicals [12].

  1. Stramandinoli-Zanicotti RT, Carvalho AL, Rebelatto CL, Sassi LM, Torres MF, Senegaglia AC, Boldrinileite LM, Correa-Dominguez A, Kuligovsky C, Brofman PR. Brazilian minipig as a large-animal model for basic research and stem cell-based tissue engineering. Characterization and in vitro differentiation of bone marrow-derived mesenchymal stem cells. J Appl Oral Sci. 2014 Jun;22(3):218-27.
  2. Ettrup KS, Glud AN, Orlowski D, Fitting LM, Meier K, Soerensen JC, Bjarkam CR, Alstrup AK. Basic surgical techniques in the Gottingen minipig: intubation, bladder catheterization, femoral vessel catheterization, and transcardial perfusion. J Vis Exp. 2011, (52):
  3. Lee JG, Park S, Bae CH, Jang WS, Lee SJ, Lee DN, Myung JK, Kim CH, Jin YW, Lee SS, Shim S. Development of a minipig model for lung injury induced by a single high-dose radiation exposure and evaluation with thoracic computed tomography. J Radiat Res. 2016, 57(3):201-209.
  4. Schuleri KH, Boyle AJ, Centola M, Amado LC, Evers R, Zimmet JM, Evers KS, Ostbye KM, Scorpio DG, Hare JM, Lardo AC. The adult Gottingen minipig as a model for chronic heart failure after myocardial infarction: focus on cardiovascular imaging and regenerative therapies. Comp Med. 2008, 58(6):568-579.
  5. Birchall MA, Ayling SM, Harley R, Murison PJ, Burt R, Mitchard L, Jones A, Macchiarini P, Stokes CR, Bailey M. Laryngeal transplantation in minipigs: early immunological outcomes. Clin Exp Immunol. 2012, 167(3):556-564.
  6. Pardo ID, Manno RA, Capobianco R, Sargeant AM, Morrison JP, Bolon B, Garman RH. Nervous System Sampling for General Toxicity and Neurotoxicity Studies in the Laboratory Minipig With Emphasis on the Gottingen Minipig. Toxicol Pathol. 2021, 49(6):1140-1163.
  7. Bollen P, Ellegaard, L. The Göttingen Minipig in Pharmacology and Toxicology. Pharmacology & Toxicology. 1997, 80 Suppl 2(2):3-4
  8. Henkel KO, Gerber T, Lenz S, Gundlach KK, Bienengräber V. Macroscopical, histological, and morphometric studies of porous bone-replacement materials in minipigs 8 months after implantation. Oral Surg Oral Med Oral Pathol Oral Radiol Endod. 2006, 102(5):606-613.
  9. Bhathena SJ, Berlin E, Johnson WA. The Minipig as a Model for the Study of Aging in Humans. In Advances in Swine in Biomedical Research. Tumbleson, Schook, Eds. Plenum Press, New York, 1996
  10. Greif G, Mrowietz C, Wendt M, Jung F, Hiebl B, Meyer-Sievers H. Differences in human and minipig platelet number, volume and activation induced by borosilicate glass beads in a modified chandler loop-system. Clin Hemorheol Microcirc. 2021, Aug 28. doi: 10.3233/CH-219201.
  11. Curtasu MV, Skou Hedemann M, Nygaard Lærke H, Bach Knudsen KE. Obesity Development and Signs of Metabolic Abnormalities in Young Gottingen Minipigs Consuming Energy Dense Diets Varying in Carbohydrate Quality. 2021, 13(5):1560.
  12. Hulse EJ, Smith SH, Wallace WA, Dorward DA, Simpson AJ, Drummond G, Clutton RE, Eddleston M. Development of a histopathology scoring system for the pulmonary complications of organophosphorus insecticide poisoning in a pig model. PLoS One. 2020, 15(10):

Comment 2

Table 1: the prevalence of Yucatan micropig is 100 % after testing one animal.  Remove the 100 % since the tested animal is only one.

Answer 2

The 100% were removed.

Comment 3

L40 : instead of the “best-known” use “ the most popular or the most known “

Answer 3

“the most known” was used.

Comment 4

L47 49 : For your statement “ The Minnesota miniature swine were the first minipigs developed in the USA in 1940s”  you can give credit or  reference to American Mini Pigs association website: https://americanminipigassociation.com/mini-pig-education/what-is-an-american-mini-pig

Answer 4

The link was added.

Comment 5

L87-91 you said “Cells of the human 293 cell line have been shown to be highly susceptible for PERV infections due to the absence of restriction factors [34]” The referred paper (reference # 34) is not related to PERV and 293. It is related with 293T and HIV-1. Please find the appropriate reference or delete the sentence.

Answer 5

Human APOBEC3G is the main inhibitor of HIV-1, but it also inhibits the replication of simian immunodeficiency virus, human T-cell lymphotropic virus, murine leukemia virus and PERV [1]. Piroozmand et al. [2] introduced APOBEC3G into 293 cells and showed that it inhibits HIV-1. We did cite this reference because the authors demonstrated by this way that APOBEC3G was not present in 293 cells. Since Lee et al. [3] did the same in a PERV system, we changed the reference.

  1. Piroozmand A, Yamamoto Y, Khamsri B, Fujita M, Uchiyama T, Adachi A. Generation and characterization of APOBEC3G-positive 293T cells for HIV-1 Vif study. J Med Invest. 2007, 54(1–2), 154–158.
  2. Jin SY, Choi HY, Kim HS, Jung YT. Human-APOBEC3G-dependent restriction of porcine endogenous retrovirus replication is mediated by cytidine deamination and inhibition of DNA strand transfer during reverse transcription. Arch Virol. 2018 Jul;163(7):1907-1914.
  3. Lee J, Choi JY, Lee HJ, Kim KC, Choi BS, Oh YK, Kim YB. Repression of porcine endogenous retrovirus infection by human APOBEC3 proteins. Biochem Biophys Res Commun. 2011 Apr 1;407(1):266-70

Comment 6

L257-258: you said “For FeLV also repeat sequences in the LTR have been identified”.  The statement is not clear or does not flow with the preceding one. It is obvious that all retroviruses’ LTRs have repeat sequences.

Answer 6

To make clearer what we wanted to say we changed the sentence as follows: For FeLV also repeat sequences inside the LTR have been identified, which may multimerize and therefore enhance the pathogenicity of the virus.

Reviewer 2 Report

The manuscript is acceptable 

Author Response

Thank you.

Reviewer 3 Report

In their manuscript “High prevalence of recombinant porcine endogenous retroviruses (PERV-A/Cs) in minipigs: a review on origin and presence“, Denner and Schuurman summarized the findings on recombination between PERV-As and PERV-Cs, a phenomenon associated with several breeds of miniature swines or minipigs. Other aspects of PERV biology and particularly the safety of xenotransplantation have been reviewed multiple times, the PERV recombination, however, has rather been neglected and requires critical assessment. The manuscript shows how much remains to be done in the field of PERV recombination and tries to draw general conclusions from the fragmentary knowledge obtained mostly in the first decade of this century. I have just a few suggestions how to make the manuscript more understandable for the readers and bring it to a wider context of retroviral recombination.

I strongly feel that papers of this type should contain a schematic representation of recombination sites in env genes. Recombination events are independent in various breeds of minipigs and arise in somatic cells, so, there should be shown the distribution of breakpoints along the retroviral sequences. From the text (lines 97-99) it looks like that there is just one breakpoint (in the env), but another breakpoint should be present upstream to the env.

It should be discused that the recombinant PERVs could be found because of their increased virulence. This is a general feature - see also the recombinant FeLVs (recombination between endo- and exogenous FeLVs), KoERVs, and ALVs (commercially important J and K ALVs are recombinnats with endogenous counterparts).

To be recombination-prone, the PERV-C copies must be transcriptionally active. Have there been any attempts to identify the recombinogenic PERV-Cs using the transcriptomic data or epigenetic features?

The authors explained the susceptibility of HEK 293 cells to PERV infection by the absence of restriction factors (line 89), but the cited paper refers to HIV-specific APOBEC3G. Is there any knowledge about the restriction factors acting against PERVs?

Author Response

Reviewer 3

In their manuscript “High prevalence of recombinant porcine endogenous retroviruses (PERV-A/Cs) in minipigs: a review on origin and presence“, Denner and Schuurman summarized the findings on recombination between PERV-As and PERV-Cs, a phenomenon associated with several breeds of miniature swines or minipigs. Other aspects of PERV biology and particularly the safety of xenotransplantation have been reviewed multiple times, the PERV recombination, however, has rather been neglected and requires critical assessment. The manuscript shows how much remains to be done in the field of PERV recombination and tries to draw general conclusions from the fragmentary knowledge obtained mostly in the first decade of this century. I have just a few suggestions how to make the manuscript more understandable for the readers and bring it to a wider context of retroviral recombination.

Comment 1

I strongly feel that papers of this type should contain a schematic representation of recombination sites in env genes. Recombination events are independent in various breeds of minipigs and arise in somatic cells, so, there should be shown the distribution of breakpoints along the retroviral sequences. From the text (lines 97-99) it looks like that there is just one breakpoint (in the env), but another breakpoint should be present upstream to the env.

Answer 1

A schematic presentation was added (Figure 1) and the text was changed accordingly:

(The figure was not copied)

Figure 1. Schematic presentation of the recombination points in PERV-A/C. A, Localization of the retroviral genes gag (group-specific antigen), encoding the core proteins, pol (polymerase), encoding the reverse transcriptase, env (envelope), encoding the envelope proteins with the receptor binding domain (RBD) and the long terminal repeats (LTR) of the integrated provirus. The recombination sites in the envelope and upstream to the env are shown. B, Examples of recombination sites in the env gene of several PERV-A/C, AY570980 PERV−A 14/220, Bartosch et al. [57], AY953542 PERV−50, Denner et al, [50], 13910, 13653, 15149, 15159, recombinant PERV-A/C from MGH miniature swines, Martin et al., [73]. A/C GM-F is a recombinant PERV-A/C isolated recently from the Göttingen minipig F [60]. The borders of the surface envelope (SU) and transmembrane envelope (TM) proteins are indicated. Blue indicated PERV−A, red PERV−B, purple is a 15 nucleotide (nt) insert in A/C GM-F, the numbers indicate the last nt of the recombination point.

Lines 99-103: Sequencing the virus isolated from the mitogen-stimulated PBMCs of the NIH minipig revealed the presence of a recombinant between PERV-A and PERV-C, one breakpoint being in the transmembrane envelope protein, the other upstream to the env [49]. Due to this recombination, the backbone PERV-C acquired the receptor binding site of the human-tropic PERV-A (Figure 1),

Comment 2

It should be discussed that the recombinant PERVs could be found because of their increased virulence. This is a general feature - see also the recombinant FeLVs (recombination between endo- and exogenous FeLVs), KoERVs, and ALVs (commercially important J and K ALVs are recombinnats with endogenous counterparts).

Answer 2

This discussion was partially present in the last version, it was now extended:

-3.2. Knowledge from the feline species, applicable to the swine species

PERV-A/C viruses were easily detected because of their increased virulence. This is a general feature of recombinant retroviruses and was well described for recombinant feline leukemia viruses (FeLV), avian leukosis viruses (ALV) and koala retrovirus (KoRV). In lymphomagenesis by murine and feline retroviruses, the acquisition of endogenous env sequences is often observed [84]. For example, feline leukemia virus A (FeLV-A) is a naturally occurring retrovirus of the cat, being horizontally transmissible from cat-to-cat. Its malignant potential depends on (i) the regulatory sequences in the LTR, (ii) the cell tropism depending on the surface envelope protein, (iii) the activation of oncogenes by insertional mutagenesis [85], and (iv) the virus load defining the severity of virus-induced immunosuppression [86]. For FeLV also repeat sequences inside the LTR have been identified, which may multimerize and therefore enhance the pathogenicity of the virus.

3.3. Knowledge from avian species, applicable to the swine species

ALV can induce various tumors and cause serious economic loss in the poultry industry. ALVs isolated from chickens were divided into six subgroups (A-J). ALV-K were isolated from chicken in China, and were identified as recombinant viruses, either as ALV-K and ALV-E recombinants, or ALV-K, ALV-E, and ALV-J multiple recombinant strains containing the U3 region of ALV-J. The viremia and viral shedding level were significantly increased, and the viruses could invade and injure the brain tissue of chickens [89]. Another example, how ALV continues evolving to obtain new genomic characters to enhance its pathogenicity is a new ALV-J strain [2]. This virus is highly pathogenic, inducing tumors in multiple organs including bone, liver, spleen, and kidney. In the viral genome recombination events with endogenous virus ev-1 sequences were identified and major recombination sites of the genome with were located in 5′ UTR-gag and 3′ UTR regions [90].

Comment 3

To be recombination-prone, the PERV-C copies must be transcriptionally active. Have there been any attempts to identify the recombinogenic PERV-Cs using the transcriptomic data or epigenetic features?

Answer 3

Unfortunately, there are no direct data on transcription and other factors. Based on the known facts, this topic was discussed:

Minipigs may be unique as they possess more copies of PERV-C sequences than some if not most other pig breeds, e.g., Large White pigs [37, 56, 70-72]. This obviously is relevant, as it has been suggested that the level of one or more PERV-C loci and their expression drives the production of PERV-A/C recombinants [58]. Genomic analysis of a MGH miniature swine capable to re-lease PERV-A/C and to infect human cells identified a PERV-C provirus in a region with homology to sequences located on chromosome 7. A PCR amplification assay with specific primers revealed that only 2 out of 5 miniature swine that are not able to transmit PERV in vitro to human or pig cells retained this locus: in contrast, this locus was found in all 5 animals able to transmit recombinant PERV-A/C to both human and pig cells and in all 7 out of 7 animals able to transmit non-recombinant PERV to pig cells [72].

Comment 4

The authors explained the susceptibility of HEK 293 cells to PERV infection by the absence of restriction factors (line 89), but the cited paper refers to HIV-specific APOBEC3G. Is there any knowledge about the restriction factors acting against PERVs?

Answer 4

Human APOBEC3G is the main inhibitor of HIV-1, but it also inhibits the replication of simian immunodeficiency virus, human T-cell lymphotropic virus, murine leukemia virus and PERV [1]. Piroozmand et al. [2] introduced APOBEC3G into 293 cells and showed that it inhibits HIV-1. We did cite this reference because the authors demonstrated by this way that APOBEC3G was not present in 293 cells. Since Lee et al. [3] did the same in a PERV system, I changed the reference.

  1. Piroozmand A, Yamamoto Y, Khamsri B, Fujita M, Uchiyama T, Adachi A. Generation and characterization of APOBEC3G-positive 293T cells for HIV-1 Vif study. J Med Invest. 2007, 54(1–2), 154–158.
  2. Jin SY, Choi HY, Kim HS, Jung YT. Human-APOBEC3G-dependent restriction of porcine endogenous retrovirus replication is mediated by cytidine deamination and inhibition of DNA strand transfer during reverse transcription. Arch Virol. 2018 Jul;163(7):1907-1914.
  3. Lee J, Choi JY, Lee HJ, Kim KC, Choi BS, Oh YK, Kim YB. Repression of porcine endogenous retrovirus infection by human APOBEC3 proteins. Biochem Biophys Res Commun. 2011 Apr 1;407(1):266-70